# Benefits of Valsartan and Amlodipine in Lipolysis through PU.1 Inhibition in Fructose-Induced Adiposity

**DOI:** 10.3390/nu14183759

**Published:** 2022-09-12

**Authors:** Chu-Lin Chou, Ching-Hao Li, Te-Chao Fang

**Affiliations:** 1Division of Nephrology, Department of Internal Medicine, School of Medicine, College of Medicine, Taipei Medical University, Taipei 110, Taiwan; 2TMU Research Center of Urology and Kidney, Taipei Medical University, Taipei 110, Taiwan; 3Division of Nephrology, Department of Internal Medicine, Shuang Ho Hospital, Taipei Medical University, New Taipei City 235, Taiwan; 4Division of Nephrology, Department of Internal Medicine, Hsin Kuo Min Hospital, Taipei Medical University, Taoyuan City 320, Taiwan; 5Department of Physiology, School of Medicine, College of Medicine, Taipei Medical University, Taipei 110, Taiwan; 6Graduate Institute of Medical Sciences, College of Medicine, Taipei Medical University, Taipei 110, Taiwan; 7Graduate Institute of Clinical Medicine, College of Medicine, Taipei Medical University, Taipei 110, Taiwan; 8Division of Nephrology, Department of Internal Medicine, Taipei Medical University Hospital, Taipei Medical University, Taipei 110, Taiwan

**Keywords:** activating transcription factor 3, adiposity, calcium channel blocker, fructose, PU.1, renin–angiotensin system blocker

## Abstract

High fructose intake has been implicated in obesity and metabolic syndrome, which are related to increased cardiovascular mortality. However, few studies have experimentally examined the role of renin–angiotensin system blockers and calcium channel blockers (CCB) in obesity. We investigated the effects of valsartan (an angiotensin II receptor blocker) and amlodipine (a CCB) on lipolysis through the potential mechanism of PU.1 inhibition. We observed that high fructose concentrations significantly increased adipose size and triglyceride, monoacylglycerol lipase, adipose triglyceride lipase, and stearoyl-CoA desaturase-1 (SCD1), activating transcription factor 3 and PU.1 levels in adipocytes in vitro. Subsequently, PU.1 inhibitor treatment was able to reduce triglyceride, SCD1, and PU.1 levels. In addition, elevated levels of triglyceride and PU.1, stimulated by a high fructose concentration, decreased with valsartan and amlodipine treatment. Overall, these findings suggest that high fructose concentrations cause triacylglycerol storage in adipocytes through PU.1-mediated activation. Furthermore, valsartan and amlodipine treatment reduced triacylglycerol storage in adipocytes by inhibiting PU.1 activation in high fructose concentrations in vitro. Thus, the benefits of valsartan and amlodipine in lipolysis may be through PU.1 inhibition in fructose-induced adiposity, and PU.1 inhibition might have a potential therapeutic role in lipolysis in fructose-induced obesity.

## 1. Introduction

Fructose intake has been linked to an increase in the epidemiological risk of metabolic syndrome [1,2]. Moreover, metabolic syndrome is epidemiologically associated with central obesity, hypertension, hyperglycemia, and hypertriglyceridemia, leading to an increase in cardiovascular disease and stroke [3,4]. In rodent models, a high-fructose (60% fructose) diet can also aggravate the activity of the renin–angiotensin system (RAS), causing insulin resistance associated with metabolic syndrome [1,2], as verified in our previous studies [3,4]. In addition, high fructose flux and metabolism in the liver lead to a rise in hepatic triglyceride accumulation [5]. Increased fructose consumption could be particularly harmful in patients with impaired glucose and lipid metabolism and increased proinflammatory cytokine expression because of the metabolic properties involved [6,7]. Thus, excessive fructose consumption has become a crucial public health concern globally, particularly in relation to obesity, nonalcoholic fatty liver disease, and type 2 diabetes [8,9].

PU.1, encoded by the Spi1 gene, is a transcription factor with multiple functions during normal and leukemogenic hematopoiesis [10]. During these processes, cellular specification is driven by primary lineage determinants, such as the transcription factor PU.1 [10]. Emerging evidence shows that PU.1 inhibition may be a potential therapeutic agent for fat degradation and its possible mechanism may be as follows [11,12,13,14]. For example, PU.1 expression in adipocytes increases significantly in genetic obesity (agouti A^vy^) [11], high-fat-diet-fed obese mice [12], and high glucose stimulation in vivo [11], which leads to insulin resistance and inflammation through the upregulation of inflammatory cytokines (TNFα, IL-1β, and IL-6) and NADPH oxidase activity [11]. Moreover, Lin et al. showed that PU.1 expression was increased only in visceral but not subcutaneous adipose tissues of obese mice, and the adipocytes were responsible for this increase in PU.1 expression [11]. In addition, activating transcription factor 3 (ATF3) functions as a transcriptional activator or repressor [15,16] and is a hub of the cellular adaptive response network, leading to induction under various inflammatory conditions. In our previous study, ATF3 deficiency in mice results in higher serum levels of triglycerides, glucose, insulin, inflammatory cytokines (ICAM-1 and TNF-α), and increasing visceral adiposity, similar to metabolic syndrome [17]. A study on *Drosophila melanogaster* was shown that ATF3 deficiency leads to chronic inflammation and metabolic disturbances [18].

The risk and prevalence of hypertriglyceridemia and intrahepatic lipid accumulation with increased fructose consumption has already been reported [8]. Postprandial hypertriglyceridemia and adiposity from excessive fructose consumption are risk factors for cardiovascular disease [8,19]. Fructose absorbed from the intestine is metabolized in the liver, where it causes lipogenesis and glycolysis, leading to triglyceride and glucose production [8,20].

Postprandial hypertriglyceridemia, a state caused by metabolic disruption resulting from excessive fructose consumption, may increase visceral adipose deposition via lipogenesis. Stearoyl-CoA desaturase-1 (SCD1) is required for the biosynthesis of triglyceride during lipogenesis [21]. Meanwhile, free fatty acids are also mobilized from adipose tissue triglycerides by the action of adipose triglyceride lipase (ATGL), hormone-sensitive lipase, and monoacylglycerol lipase (MGL), which are involved in adipose tissue lipolysis, including the hydrolysis process of triglyceride, fatty acids, and glycerol [22,23,24].

Visceral adiposity contributes to hepatic triglyceride accumulation and hepatic insulin resistance by increasing the portal delivery of free fatty acids to the liver [25]. By contrast, after fasting or the high energy demands of physical activity, lipolysis is vital for supplying fatty acids and glycerol as energy to tissues [23,26]. In addition to providing energy, intermediate and end products in the adipose lipolysis process are able to regulate the metabolic process in nonadipose tissue [27]. Changes in the lipolysis process have frequently been associated with lipodystrophy, hyperlipidemia, insulin resistance, type 2 diabetes, depression, and cancer [23,27]. Furthermore, PU.1 reportedly expresses itself in adipocytes as an insulin resistance and inflammation factor, which increases significantly with high glucose stimulation in vivo [11]. However, few studies have focused on PU.1 in lipolysis.

Our previous studies with fructose-fed rodents with metabolic syndrome revealed that RAS blockers could ameliorate hypertriglyceridemia [3] and visceral adiposity [28], as well as improve aortic endothelial functions and reduce oxidative stress [4]. In addition, as previously reported, calcium channel blockers (CCB) attenuate white adipose tissue dysfunction and increase adipocyte differentiation in type 2 diabetic KK-A(y) mice [29]. However, few studies have experimentally examined the role of RAS blockers and CCB in adiposity. Therefore, we investigated the effects of RAS blockers and CCB on lipolysis through the potential mechanism of PU.1 inhibition.

## 2. Materials and Methods

### 2.1. Cell Culture and Adipocyte Differentiation

Mouse 3T3-L1 was purchased from American Type Culture Collection (ATCC, Manassas, VA, USA). The cells were routinely nourished in low-glucose Dulbecco’s modified Eagle’s medium, supplemented with 10% (*v*/*v*) fetal bovine serum, 1% penicillin/streptomycin mixture, and glutamate, at 37 °C in a humidified atmosphere containing 5% CO_2_. The complete medium was refreshed every 2 days.

For adipocyte differentiation, 3T3-L1 cells were seeded at 4 × 10^4^ per cm^2^ (See the Appendix A). Two days later (70–80% confluency), the complete medium was replaced with differentiation medium A (MA: complete medium supplemented with 50 mg/L insulin, 0.5 mM of 3-isobutyl-1-methylxanthine, and 1 μM of dexamethasone) for 2 days (denoted as Day 0). The cells were then incubated with differentiation medium B (MB: complete medium supplemented with 50 mg/L insulin). The MB was refreshed every 2 days. In general, the adipocyte differentiation could be identified from Day 3, and an apparent complete differentiation can be achieved between Day 7 and Day 10 [30]. In the control treatment, 3T3-L1 cells were differentiated with MB alone, whereas in the testing groups, 3T3-L1 cells were incubated with MB in the presence of fructose at various concentrations (1, 2, and 4 mg/mL). Cells in control and testing groups were collected after the same incubation period, and samples were collected at Day 7 of the experimental protocol. The RAS/or CCB blocker or the PU1 inhibitor was supplied in MB medium throughout the differentiation period.

### 2.2. Oil Red O Staining

Preadipocyte differentiation was imaged at Day 3 and Day 7. In addition, the accumulation of lipid droplets was quantified by Oil Red O (ORO) staining. Briefly, the differentiated adipocytes were washed twice with phosphate-buffered saline and fixed for 1 h in 10% formalin. The cells were then immersed briefly in 60% isopropanol and air dried. After a 30-min incubation with the ORO working solution, the cells were washed several times in distilled water to remove the unbinding dye. The accumulation of lipid droplets was determined by microscopy. For quantitative measurements, the ORO staining was eluted with 100% isopropanol (*v*/*v*), and the optical absorbance was measured at 595 nm [30], which was based on the quantitative method of the previous studies [30,31,32].

### 2.3. Western Blotting

Whole-protein lysate of differentiated adipocytes was extracted using a RIPA lysis buffer (containing a protease and phosphatase inhibitor cocktail). Western blotting was performed as described in [33]. Briefly, 40 μg of whole lysate was pipetted into a 12% separation gel and then subjected to constant voltage (90 V) SDS-PAGE. After separation, the proteins inside the gels were transferred to polyvinylidene fluoride (PVDF) membranes with a constant current (400 mA). The PVDF membranes were then washed with tris-buffered saline (with 0.05% Tween-20) containing 5% bovine serum albumin. Subsequently, the membranes were covered with primary and horseradish peroxidase (HRP)-conjugated secondary antibodies, respectively. The primary antibodies used in this study at the following dilutions in phosphate-buffered saline with Tween 20 (PBST): anti-activating transcription factor 3, 1/1000 (ATF3; GTX30069, GeneTex, Irvine, CA, USA), anti-PU.1, 1/1000 (GTX101581, GeneTex), and anti-β-actin, 1/10,000 (Sigma-Aldrich, St. Louis, MO, USA). Enhanced chemiluminescence was used for the visualization of the blotting bands, which were captured using a BioSpectrum AC imaging system (UVP, Upland, CA, USA). The intensity of the blotting bands was quantified using Gel-Pro analyzer software (version 4.0, Media Cybernetics, Rockville, MD, USA).

### 2.4. Reverse Transcription-Quantitative Polymerase Chain Reaction Assay

Total RNA was extracted using the Trizol reagent (Thermo Fisher Scientific, Waltham, MA, USA) according to the manufacturer’s protocol. cDNA was synthesized using the MMLV RT-Script kit (BioGenesis, Taipei, Taiwan), and a quantitative polymerase chain reaction (qPCR) was performed using OmicsGreen qPCR Master Mix with ROX dye (Omicsbio, New Taipei City, Taiwan) according to the manufacturer’s protocol. Three-step PCR cycling was conducted using LightCycler Nano software (Roche Molecular Systems, Almere, The Netherlands) with primer sequences [34] and the following settings: 95 °C for 2 min, followed by 40 cycles at 95 °C for 20 s, 60 °C for 20 s, and 72 °C for 40 s, and then a final extension step of 95 °C for 15 s, 60 °C for 1 min, 95 °C for 15 s, and 60 °C for 15 s. Glyceraldehyde-3-phosphate dehydrogenase was used as an internal control for normalization. Gene expression was calculated using the 2^−ΔΔCq^ method [35].

### 2.5. Hematoxylin and Eosin Staining and the Quantification of Lipid Droplet Size

Fat tissues of mice fed with normal diet and 8-week high-fructose diet were obtained from our previous study (Protocol Number: LAC-2016-0093), and all experimental procedures were approved by the Institutional Animal Care and Use Committee of Taipei Medical University (Protocol Number: LAC-2016-0093) and were in strict accordance with the recommendations of the Guide for the Care and Use of Laboratory Animals set by the National Institutes of Health [17]. Fat tissues were embedded in paraffin and then they were cut into 0.5-μm-thick slices. In brief, control group, comprising wild-type mice that were fed a standard chow diet for 8 weeks; and fructose group, comprising wild-type mice that were fed a high-fructose (60% fructose) diet for 8 weeks. To investigate the histological changes, hematoxylin and eosin (HE; Bio-Check Laboratories, New Taipei City, Taiwan) staining was conducted in accordance with the manufacturer’s protocol. HE-stained slices of fat tissue were observed under a microscope (Olympus IX70, Tokyo, Japan) equipped with ViewPoint Virtual Slide Viewing Software, (PreciPoint, Freising, Germany), and the size of the lipid droplets was manually traced using the software’s scaling tool. The average lipid droplet diameter was calculated on the basis of >50 lipid droplets per image [33].

### 2.6. Immunohistochemistry Staining

To observe the expression pattern of ATF3 and PU.1, the paraffin-embedded slices of fat tissue were deparaffined, antigen retrieval was performed, and the antigens were incubated with the antibodies specific to ATF3 (GTX02578, GeneTex) at a dilution of 1/200 and PU.1 (DF13270, Affinity Biosciences, OH, USA) at a dilution of 1/50, respectively. Following routine procedures regarding secondary antibody incubation, washing, and HRP-DAB (3,3′-diaminobenzidine)-based chromogenicity, the chromogenic reactivity was detected using the light microscope, as described previously [33]. Images were acquired, and the immunoreactivity was quantified using ImageJ 1.51 software (NIH, Bethesda, MD, USA).

### 2.7. Statistical Analyses

All the results are expressed as mean ± standard deviation (SD). A one-way analysis of variance (ANOVA), followed by the Newman–Keuls post hoc test or one-way repeated-measures ANOVA, was performed for group comparisons. Additionally, a two-way ANOVA (first factor: treatment group; second factor: time period), with the two drug treatments as the factors, was performed to compare the groups. A Student *t* test was performed for the unpaired data when appropriate. *p* values significant at the <0.05, <0.01, and <0.001 levels are reported.

## 3. Results

### 3.1. Effects of Different Fructose Concentrations on Adipocyte Size and Triglyceride, ATF3, PU.1, MGL, ATGL, and SCD1 Levels in Adipocytes In Vitro

We examined the effects of high fructose concentrations on lipogenesis and lipolysis in vitro treated by DB2313 (PU.1 inhibitor), amlodipine (a CCB), valsartan (an angiotensin II receptor blocker, ARB), and untreated. Figure 1 presents the effects of different fructose concentrations on triglyceride levels by measuring ORO staining and adipose size in adipocytes in vitro. Through this method, we discovered that an increased fructose concentration stimulated adipocytes, causing adipose enlargement and triglyceride accumulation. Figure 2 indicates the effects of different fructose concentrations on ATF3 and PU.1 protein levels, ATF3, PU.1, MGL, ATGL, and SCD1 mRNA levels, and the immunohistochemistry (IHC) intensity of ATF3 and PU.1 in adipocytes in vitro. An increased fructose concentration gradually raised ATF3 and PU.1 protein levels and of PU.1 mRNA levels but reduced ATF3 mRNA levels. Moreover, an increased fructose concentration of 4 mg/mL augmented SCD1, ATGL, and MGL mRNA levels. In IHC staining, a 4 mg/mL fructose concentration enhanced ATF3 and PU.1 expression in adipocytes in vitro.

### 3.2. Effects of a PU.1 Inhibitor on Adipocyte Triglyceride, ATF3, PU.1, MGL, ATGL, and SCD1 in a 4 mg/mL Fructose Concentration In Vitro

The effects of the fructose concentration (4 mg/mL) with and without a PU.1 inhibitor (DB2313, 5 nM) on triglyceride levels by measuring ORO staining, ATF3, and PU.1 protein levels, and ATF3, PU.1, MGL, ATGL, and SCD1 mRNA levels in adipocytes in vitro are illustrated in Figure 3. Triglyceride, ATF3 protein, and PU.1 protein levels increased as a result of fructose stimulation and subsequently decreased after treatment with the PU.1 inhibitor. Moreover, PU.1 mRNA levels increased with a relatively high fructose concentration of 4 mg/mL and then decreased following PU.1 inhibitor treatment; however, the ATF3 mRNA levels were inversely increased by the PU.1 inhibitor. The levels of MGL, ATGL, and SCD1 induced by the 4 mg/mL fructose concentration were significantly higher than in the control without fructose. The PU.1 inhibitor increased MGL and ATGL mRNA levels and reduced SCD1 mRNA levels. Appendix A showed the protein level of ATGL was induced, whereas the SCD1 was decreased, during adipogenesis in responded to the treatment of fructose. Basically, the protein level of ATGL was coincided with its mRNA pattern. However, the translational level of SCD1 was obviously repressed upon the maturation of adipocyte. The PU.1 inhibitor reduced SCD1 protein level, but unaltered the ATGL protein level.

### 3.3. Effects of ARB and CCB on Adipocyte Triglyceride, ATF3, PU.1, MGL, ATGL, and SCD1 Levels in a 4 mg/mL Fructose Concentration In Vitro

Figure 4 depicts the effects of a 4 mg/mL M fructose concentration with and without amlodipine (10 μM) and valsartan (10 μM) on triglyceride levels by measuring ORO staining, ATF3 and PU.1 protein levels, and ATF3, PU.1, MGL, ATGL, and SCD1 mRNA levels in adipocytes in vitro. The triglyceride levels increased by the high fructose concentration were reduced by the amlodipine and valsartan treatment. The high fructose concentration also increased ATF3 and PU.1 protein levels; subsequently, only the PU.1 protein levels decreased as a result of the valsartan and amlodipine treatment, although the reduction caused by valsartan was nonsignificant. Moreover, the valsartan and amlodipine treatment significantly reduced the PU.1 mRNA levels. The increased MGL, ATGL, and SCD1 mRNA levels as a result of the 4 mg/mL fructose concentration remained unaffected by the valsartan treatment. Additionally, although the valsartan treatment resulted in a decreasing trend in SCD1 mRNA levels, the reduction was nonsignificant. Appendix A showed Co-incubation with valsartan (10 μM) and amlodipine (10 μM) sustained the high-expressed ATGL protein level and the repression of SCD1 protein translation.

## 4. Discussion

The key findings of our study are summarized in this section. First, high fructose concentrations increased adipocyte size and triglyceride content in adipocytes in vitro. Second, the high fructose concentration significantly increased MGL, ATGL, and SCD1 levels; subsequently, SCD1 levels decreased following PU.1 inhibitor treatment in vitro. Third, the high fructose concentration increased adipose triglyceride, ATF3, and PU.1 levels, and triglyceride and PU.1 levels decreased after PU.1 inhibitor treatment. Fourth, increased triglyceride and PU.1 levels in adipocytes treated with high fructose concentrations were reduced with the amlodipine and valsartan treatment. Overall, these findings suggest that high fructose concentrations cause lipogenesis through PU.1 activation, and amlodipine and valsartan treatment activate lipolysis through a potential mechanism that inhibits PU.1 activation.

High fructose intake contributes to the development of obesity [8]. Fructose is metabolized in the intestine and transported into the liver, where approximately two-thirds of the fructose intake can be metabolized; the remaining intake is metabolized by other tissues, including adipose tissue expressed as the fructose transporter GLUT5 [36]. Furthermore, postprandial hypertriglyceridemia and visceral adipose deposition are metabolic disturbances resulting from excessive fructose consumption. Our study revealed that fructose stimulation increased MGL, ATGL, and SCD1 levels, and that SCD1 could subsequently be reduced by the PU.1 inhibitor. In addition, through the action of ATGL and MGL in the adipose tissue, adipose tissue triglycerides were metabolized into free fatty acids, which are involved in adipose tissue lipolysis.

In our study, the data revealed that fructose stimulation increased the size of adipose tissue and the lipid droplets, as well as increased ATF3 and PU.1 protein levels, which were subsequently lowered by the PU.1 inhibitor. Ruan et al. demonstrated the inhibition of PU.1 in the adipogenesis of ovine primary preadipocytes by limiting CCAAT-enhancer-binding protein-β [13]. In adipocyte-specific PU.1-knockout mice, peroxisome proliferator–activated receptor-gamma (PPARγ) is more active when PU.1 expression is reduced in adipocytes [12], indicating that increased PU.1 modifies the adipocyte PPARγ in adipocytes. PPARγ is highly expressed in adipose tissue, and it can be activated to alter the adipocyte phenotype and upregulate genes involved in fatty acid metabolism and triglyceride storage [14]. For example, PPARγ-activating ligands improve the function of fatty tissues, such as the expression of adiponectin and the reduction in nonesterified fatty acid intake in plasma [14]. This evidence suggests that fructose-stimulated PU.1 may be a mediator that contributes in part to the regulation of adipogenesis in adipose tissue. Thus, PU.1 could be a potential therapeutic agent for fat degradation through improving oxidative stress and regulating PPARγ-mediated metabolic processes.

Whether ATF3 affects adipose lipid processing is still being researched. In this in vitro study, we evaluated the association between ATF3 and lipolysis in adipose cells with and without high fructose stimulation. Our previous study demonstrated that the loss of ATF3 in mice was accompanied by elevated serum levels of glucose, insulin, triglycerides, inflammatory cytokines (TNF-α and ICAM-1), and larger visceral adiposity compared with wild-type mice [17]. This resembled metabolic syndrome and indicated that ATF3 could be involved in regulating lipogenic properties. That study further observed that high fructose stimulated an increased protein response in ATF3 and PU.1 and enhanced lipid droplet sizes with cumulative MGL, ATGL, and SCD1 levels, which were subsequently reduced by using the PU.1 inhibitor. Furthermore, Suganami et al.’s study provided evidence that ATF3, induced in obese adipose tissue, acts as a transcriptional repressor of the negative feedback mechanism that attenuates saturated fatty acid/toll-like receptor 4 signaling and macrophage activation in obese adipose tissue [37]. Moreover, our colleagues found that a synthetic ATF3 inducer, ST32da, promoted ATF3 expression to downregulate adipokine genes, increasing white adipose tissue browning and reducing lipogenesis in ATF3-knockout mice with a high-fat diet [34].

In this study, we found that there is a discrepancy between ATF3 protein and mRNA levels. As mRNA is eventually translated into protein, it is usually assumed that there is some sort of correlation between the levels of mRNA and protein. However, poor correlations between the levels of mRNA and protein are common, and this could be achieved by many factors. For example, the half-lives of mRNA and protein may differ. In addition, the modification of mRNA (e.g., methylation) might change the translational efficiency of mRNA. Additionally, the increased protein level may down-regulate mRNA synthesis and up-regulate mRNA degradation. For example, Zhu and his colleagues found the ATF3 protein in colorectal cancer patients was statistically up-regulated, but the ATF3 mRNA level was without significant change [38], suggesting a discrepancy between ATF3 protein and mRNA levels. The overexpressing ATF3 (by plasmid) inhibited adipocyte differentiation of 3T3-L1 cells [39,40]. However, a synthetic ATF3 inducer ST32da not only promoted ATF3 expression, but also induced adipocyte browning, suggesting the need for ATF3 during glucose metabolism and lipogenesis [34]. Moreover, ATF3 can have opposite effects on different types of cells. The functions of ATF3 might depend on its transcriptional milieu, thus the levels of protein and mRNA of ATF3 were investigated in this study.

Our study is the first to reveal that amlodipine and valsartan treatment could reduce fat droplets and PU.1 protein levels in adipocytes. Previous studies have reported the beneficial role of amlodipine and valsartan in improving lipoprotein oxidation, increasing superoxide dismutase activity and improving the effects of atherosclerosis in experimental and human studies [41,42,43,44,45,46,47,48,49]. In particular, local angiotensin II plays a key role in promoting the adipogenic differentiation of mesenchymal stem cells from human fat tissue through type 2 angiotensin receptors [50]. Furthermore, the activation of RAS by angiotensin II treatment increases endoplasmic reticulum stress and causes the inflammation of adipocytes through angiotensin II receptor type 1 [51]. ARBs have therefore been reported to have antioxidant and anti-inflammatory effects and result in improved fat-droplet size, as well as the differentiation and dysregulation of adipocytokines in culture adipocytes and mice [52,53,54,55]. In the future, the association between PU.1 inhibitors and RAS-induced paracrine effects merits investigation. In addition, nifedipine, a CCB, attenuates white adipose tissue dysfunction in type 2 diabetic KK-A(y) mice, demonstrating that nifedipine increases adipocyte numbers and the expression of PPARγ and adipocyte fatty acid-binding proteins related to adipocyte differentiation [29]. Furthermore, in essential hypertensive patients, Harano et al. observed that the responses of ketone bodies during insulin sensitivity tests at 30 min improved after amlodipine treatment, which reflects the effect of insulin on lipolysis in adipose tissue and hepatic fatty acid oxidation [44]. As for the role of ARBs and CCBs in adipocyte PU.1 inhibition, further studies are required to identify the mechanism in detail.

Our study demonstrated that high fructose stimulation increased adipose size and triglyceride, MGL, ATGL, SCD1, ATF3, and PU.1 levels in adipocytes in vitro; subsequently, PU.1 inhibitor treatment reduced triglyceride, SCD1, and PU.1. In addition, we revealed that increased triglyceride, and PU.1 levels in adipocytes stimulated by relatively high fructose concentrations were reduced by treatment with amlodipine and valsartan. Amlodipine increased fructose-induced SCD1 mRNA levels, which may be because amlodipine partially inhibited PU.1 and also had the other signaling pathways on lipogenesis independent of PU.1 signaling. These results indicate that high fructose concentrations cause triacylglycerol storage in adipocyte droplets through PU.1 activation. Furthermore, amlodipine and valsartan treatment can improve triacylglycerol storage in adipocyte droplets by inhibiting PU.1 activation. Thus, the benefits of valsartan and amlodipine in lipolysis may be through PU.1 inhibition in fructose-induced adiposity, and PU.1 inhibition might have a potential therapeutic role in lipolysis in fructose-induced obesity.

## Figures and Tables

**Figure 1 nutrients-14-03759-f001:**
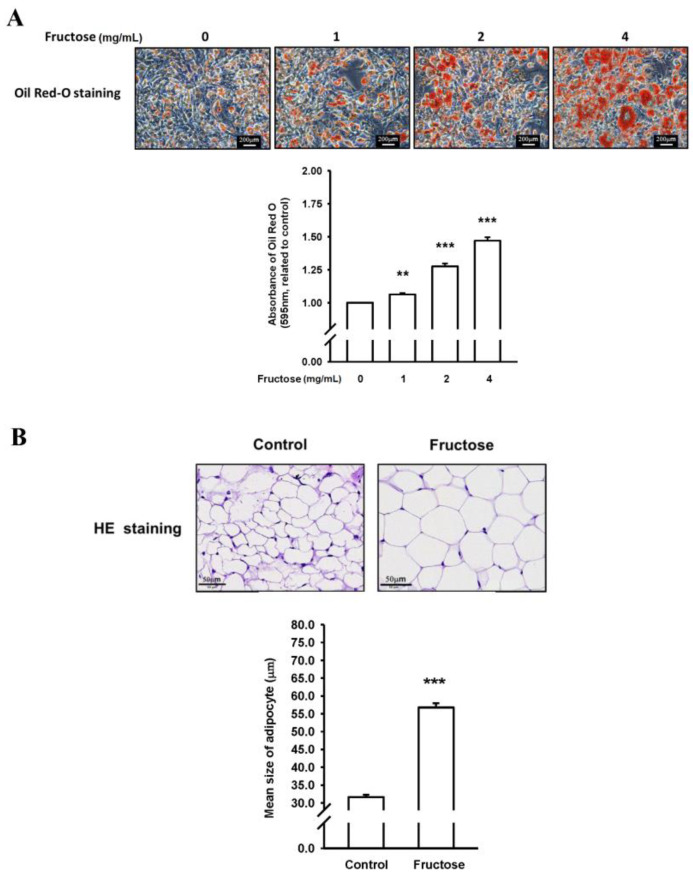
Effects of different fructose concentrations on (**A**) triglyceride levels by measuring Oil Red O (ORO) staining in adipocytes in vitro and (**B**) adipose size obtained from our previous study [17]. N = 6, the results from 6 independent experiments in cell culture experiments and from 6 mice in tissue used study. Values are presented as mean ± standard deviation (SD). **, and *** denote *p* < 0.01, and <0.001 vs. the control groups at Day 7 of the experimental protocol, respectively.

**Figure 2 nutrients-14-03759-f002:**
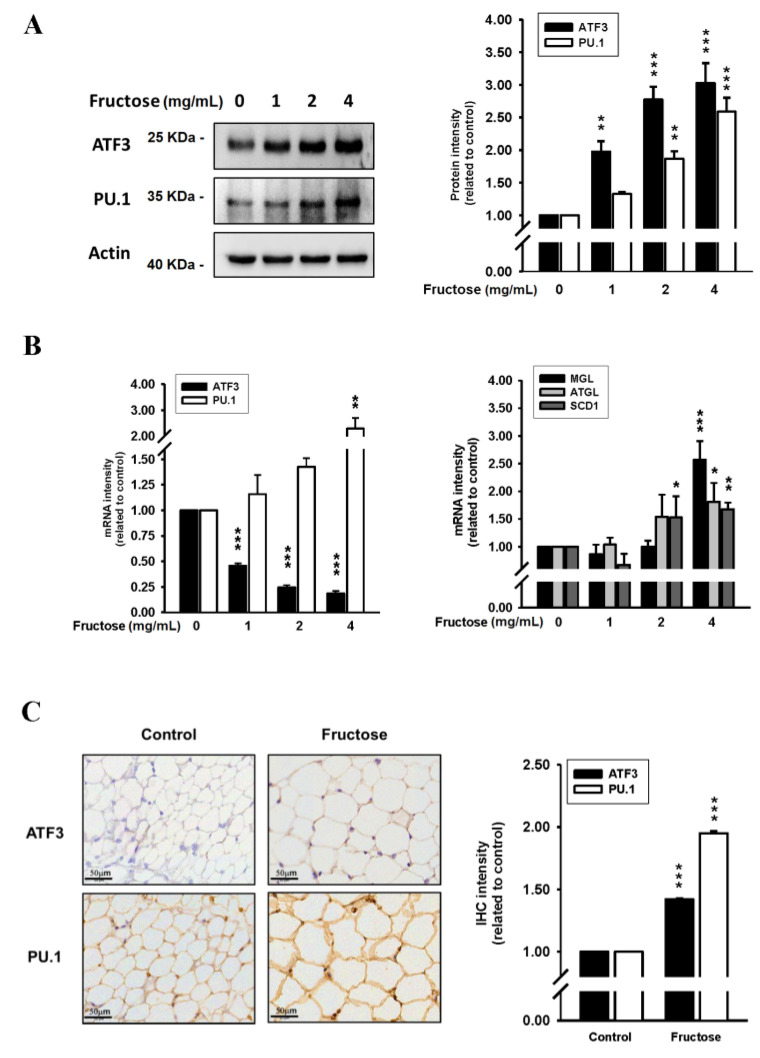
Effects of different fructose concentrations on (**A**) ATF3 and PU.1 protein levels in adipocytes in vitro, (**B**) ATF3, PU.1, MGL, ATGL, and SCD1 mRNA levels in adipocytes in vitro, and (**C**) the immunohistochemistry intensity of ATF3 and PU.1 obtained from our previous study [17]. N = 6, the results from 6 independent experiments in cell culture experiments and from 6 mice in tissue used study. Values are presented as mean ± SD. *, **, and *** denote *p* < 0.05, <0.01, and <0.001 vs. the control groups at Day 7 of the experimental protocol, respectively.

**Figure 3 nutrients-14-03759-f003:**
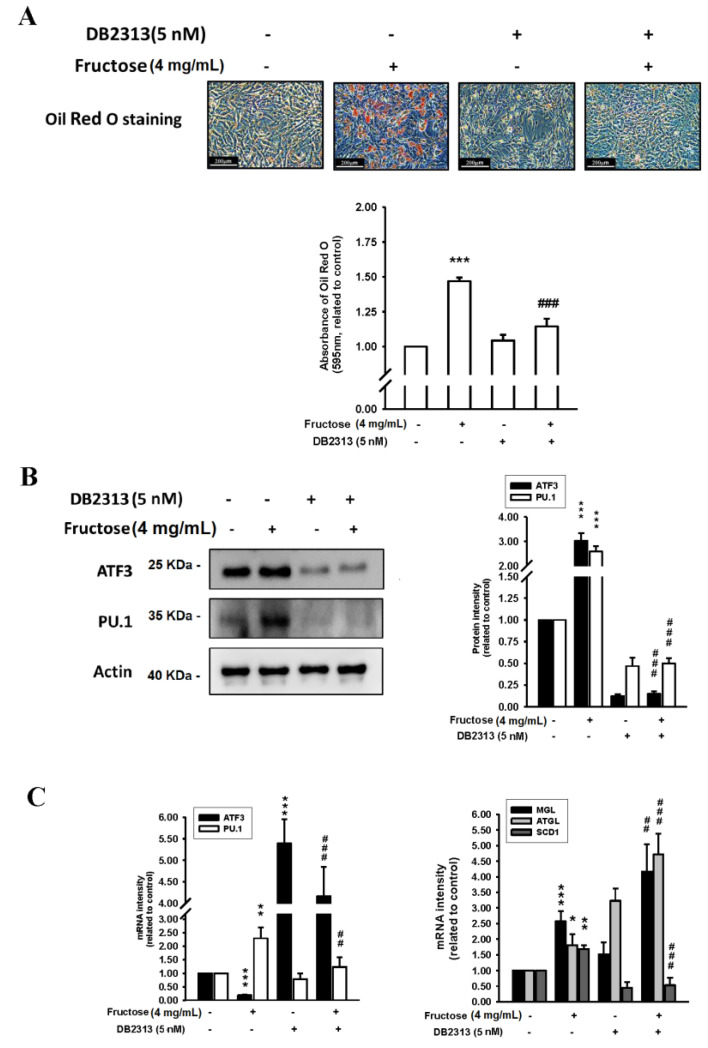
Effects of fructose concentration (4 mg/mL) with and without a PU.1 inhibitor (DB2313, 5 nM) on (**A**) triglyceride levels by measuring ORO staining, (**B**) ATF3 and PU.1 protein levels, and (**C**) ATF3, PU.1, MGL, ATGL, and SCD1 mRNA levels in adipocytes in vitro. N = 6, the results from 6 independent experiments in cell culture experiments. Values are presented as mean ± SD. *, **, and *** denote *p* < 0.05, <0.01, and <0.001 vs. the control groups at Day 7 of the experimental protocol, respectively. ##, and ### denote *p* <0.01, and <0.001 vs. the 4 mg/mL fructose group at Day 7 of the experimental protocol, respectively.

**Figure 4 nutrients-14-03759-f004:**
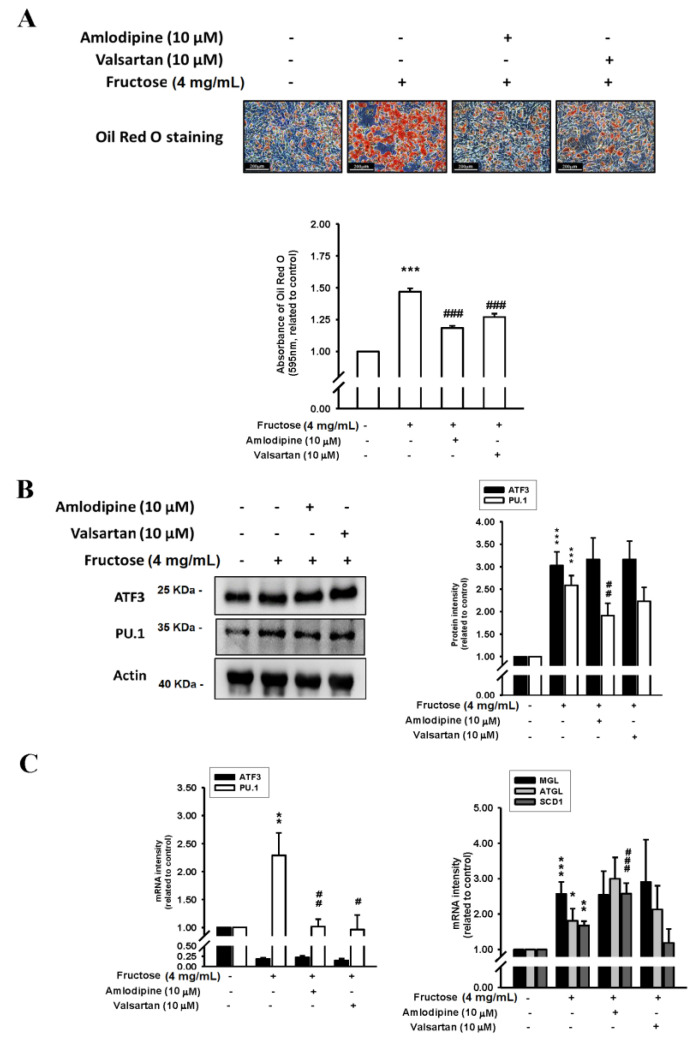
Effects of fructose concentration (4 mg/mL) with and without amlodipine (10 μM) and valsartan (10 μM) on (**A**) triglyceride levels by measuring ORO staining, (**B**) ATF3 and PU.1 protein levels, and (**C**) ATF3, PU.1, MGL, ATGL, and SCD1 mRNA levels in adipocytes in vitro. N = 6. the results from 6 independent experiments in cell culture experiments. Values are presented as mean ± SD. *, **, and *** denote *p* < 0.05, <0.01, and <0.001 vs. the control groups at Day 7 of the experimental protocol, respectively. #, ##, and ### denote *p* < 0.05, <0.01, and <0.001 vs. the 4 mg/mL fructose group at Day 7 of the experimental protocol, respectively.

## Data Availability

Data is contained within the article.

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
