# Peer review of "Benefits of Valsartan and Amlodipine in Lipolysis through PU.1 Inhibition in Fructose-Induced Adiposity"

_nutrients, 2022, doi:10.3390/nu14183759_

Round 1

Reviewer 1 Report

In this manuscript, authors examined the effect of renin–angiotensin system blocker and calcium channel blocker on adipose size, triglyceride, adipogenesis and lipolysis related genes. Several major concerns regarding the experimental design and results need to be addressed.

Introduction

Please define ATF3.  What is the role of ATF3 in fructose related metabolism? Please state this in the Introduction?

Methods

1. Section 2.1. Were 3T3-L1 cells in control and testing group collected after same incubation period or  after complete differentiation but at different incubation periods? Please state clearly the differentiation days for both control and testing groups.

Results

1. As noted in Figures legends, “*, **, and *** denote P < 0.05, < 0.01, and < 0.001 vs. the control groups at the corresponding time points, respectively”.  Were cells in control and testing groups collected at different time points? Please indicate the time points here and/or Methods section.  

2. In this MS, the major outcomes include PU.1 as well as ATF3, MGL, ATGL and SCD1.  Fructose increased both protein and mRNA levels of PU.1 while “there is a discrepancy between ATF3 protein and mRNA 300 levels (line 300 and Figures 2,3 &4). For MGL, ATGL and SCD1, authors showed the mRNA levels only. Did authors measured protein levels/activities changes of MGL, ATGL and SCD1? 

3. In Figure 3C, PU.1 blocker reduced triglycerides, decreased PU.1 and reduced fructose induced SCD1 mRNA levels. However, as shown in Figure 4, Amlodipine addition reduced triglycerides, decreased PU.1, however, further increased fructose induced SCD1 mRNA levels.  Is this due to the differences of inhibition between PU.1 blocker and Amlodipine?

Reviewer 2 Report

This article by Chou et al. is an excellent study with very interesting results. High fructose consumption is evident to lead to chronic metabolic heath problems including obesity, diabetes, hepatic and cardiovascular abnormalities. The authors demonstrated a range of high fructose induced metabolic derangements in vitro as well as in adipose tissue derived from high fructose fed mice. Importantly, the study provided mechanistic insights showing that the high fructose induced effects are mediated by a transcription factor called PU.1 and inhibition of PU.1 can reverse the effects of high fructose consumption.  They further demonstrated that beneficial effects of valsartan (a renin-angiotensin receptor blocker) and amlodipine ( a calcium channel blocker) are potentially through the inhibition of PU.1 activity in adipocytes. 

There are only a few minor issues as mentioned below:

1.  It would be better to mention the dilution factors of primary antibodies used for western blotting and immunohistochemistry. 

2. For all the western blot images, the author should label the gel with the size of the protein (molecular weight).

3. Page 4, line 151-152, the authors mentioned using fat tissue from mice from a previous study. Here the author should report the ethical approval number of that study. Similarly, on page 11, line 351-351, please provide the IRB approval number. 

4. I am not clear about the n=6 in the figures. Is it the results from 6 independent experiments or tissues used from 6 mice? 

5. The inconsistency in mRNA and protein levels for ATF3. But the authors have provided explanation for this in the discussion. 

6. Page 9, line 272, there is a typo "ovine". Should it be "bovine"? Please correct. 
